# A Three-Dimensional Vibration Theory for Ultralight Cellular Sandwich Plates Subjected to Linearly Varying In-Plane Distributed Loads

**DOI:** 10.3390/ma16114086

**Published:** 2023-05-31

**Authors:** Fei-Hao Li, Bin Han, Ai-Hua Zhang, Kai Liu, Ying Wang, Tian-Jian Lu

**Affiliations:** 1The Institute of Xi’an Aerospace Solid Propulsion Technology, Xi’an 710025, China; xjtulfhao@163.com (F.-H.L.);; 2National Key Laboratory of Solid Rocket Propulsion, Xi’an 710025, China; 3School of Mechanical Engineering, Xi’an Jiaotong University, Xi’an 710049, China; 4State Key Laboratory of Mechanics and Control of Mechanical Structures, Nanjing University of Aeronautics and Astronautics, Nanjing 210016, China; 5Nanjing Center for Multifunctional Lightweight Materials and Structures, Nanjing University of Aeronautics and Astronautics, Nanjing 210016, China

**Keywords:** cellular sandwich plate, fundamental frequency, linearly varying in-plane distributed load, zigzag shear theory

## Abstract

Thin structural elements such as large-scale covering plates of aerospace protection structures and vertical stabilizers of aircraft are strongly influenced by gravity (and/or acceleration); thus, exploring how the mechanical behaviors of such structures are affected by gravitational field is necessary. Built upon a zigzag displacement model, this study establishes a three-dimensional vibration theory for ultralight cellular-cored sandwich plates subjected to linearly varying in-plane distributed loads (due to, e.g., hyper gravity or acceleration), with the cross-section rotation angle induced by face sheet shearing accounted for. For selected boundary conditions, the theory enables quantifying the influence of core type (e.g., close-celled metal foams, triangular corrugated metal plates, and metal hexagonal honeycombs) on fundamental frequencies of the sandwich plates. For validation, three-dimensional finite element simulations are carried out, with good agreement achieved between theoretical predictions and simulation results. The validated theory is subsequently employed to evaluate how the geometric parameters of metal sandwich core and the mixture of metal cores and composite face sheets influence the fundamental frequencies. Triangular corrugated sandwich plate possesses the highest fundamental frequency, irrespective of boundary conditions. For each type of sandwich plate considered, the presence of in-plane distributed loads significantly affects its fundamental frequencies and modal shapes.

## 1. Introduction

With the development of modern flight vehicles, studies on advanced materials, topological structures, and biological processes in hyper gravity environments have evolved into an important research hotspot [1,2,3]. It has been demonstrated that gravity has noticeable effects on the precision of positioning systems, inertial navigators, and guided systems as well as the mechanical performance of thin structural elements applicated in aerospace vehicles [4,5,6]. Typically, such thin structural elements include large-scale covering shells/plates of aerospace protection structures, vertical stabilizers of accelerated aircraft or missiles, and structural components of specific systems strongly influenced by varying gravity [7,8]. These structures, relatively thin, are subjected to hyper gravity as the flight vehicle accelerates. It is thus necessary to explore how the mechanical behaviors (e.g., buckling and vibration) of thin-walled structures are affected by gravitational field.

As an elemental thin structure component, the mechanical performance of a monolithic plate in gravitational field has been extensively studied. At the early stage, the stability and vibration of a rectangular plate subjected to self-weight, or acceleration-induced body force, were characterized by treating the body force as linearly varying in-plane distributed loads (IPDLs) [9,10,11]. Afterwards, the elastic stability and vibration of a standing monolithic plate with different boundary conditions (BCs) were investigated by applying either the classical or the Reissner–Mindlin plate theory [12,13,14,15,16,17]. It has been demonstrated that IPDL significantly affects the mechanical properties of a monolithic plate. For instance, IPDL affects the potential energy of the plate, thus also changing its stability and vibration behaviors. The critical buckling top load decreases with increasing IPDL, while the fundamental frequency can be either greater or less than that of a no-load plate, depending on the type of BCs and the direction of accelerations.

In lieu of monolithic plates, sandwich structures are widely utilized in aircraft, aerospace vehicles, satellites, missiles, and the like. In particular, ultralight sandwich plates with thin face sheets and periodic lattice truss cores, for example, triangular corrugated (TCOR) plates and hexagonal honeycombs (HHON), receive increasing attention due to superior load-bearing capability and additional multifunctional attributes such as energy absorption, active cooling, and noise attenuation [18,19,20]. For instance, it has been demonstrated that TCOR and HHON sandwich plates possess excellent bending strength, blast resistance, and impact energy absorption [21,22]. However, while the influence of hyper gravity or acceleration on either vibration or stability of sandwich plates deserves much attention as in the case of monolithic plates, only the present authors managed to investigate how a corrugated sandwich plate would buckle when subjected to distributed body force under a variety of BCs [23]. At present, the vibration behaviors of ultralight cellular-cored sandwich plates subjected to IPDLs remain elusive.

This study aims to establish a theory to analyze the vibration performance of ultralight cellular-cored sandwich plates subjected to linearly varying IPDLs, with the effects of BCs duly accounted for. In previous theoretical analysis of sandwich plates, they are often regarded as a special case of laminated structure [24]. Additionally, there are four main kinds of theoretical displacement model: three-dimensional elasticity, equivalent single layer, layer wise, and zigzag [25,26]. The three-dimensional elasticity theory regards the laminated structure as a generalized three-dimensional solid model, and does not consider the special layered configuration of the laminates, and the calculation cost is generally relatively high. At present, the equivalent single-layer model has classical theories and various high-order theories, but it ignores the continuity conditions of interlayer displacement and transverse shear stress. Layer wise model and zigzag model can overcome the limitations of the previous two theoretical models by independently modeling the displacement field of each single layer, and achieve a balance between computational accuracy and computational efficiency. However, the unknown displacement functions of the layer wise model increase with the increase of the number of laminate layers. The zigzag model is actually a superposition form of the equivalent single-layer model and interlayer continuous terms. The number of unknown variables is certain, and the calculation amount is less than the layer wise model [27,28]. To sum up, this paper intends to use the zigzag displacement model. Zigzag theory was pioneered by Lekhnitskii for multilayered beams, and Di Sciuva is the first to apply the zigzag model to the vibration analysis of the sandwich structure [29,30]. The bending, buckling, and vibration of simply supported multilayer anisotropic plates are analyzed, and the in-plane displacement is distributed linearly along the thickness of the structure under the condition of continuity between layers. The defect of Di Sciuva’s model is that it cannot be used to analyze the global response of the structure under clamped boundary conditions. Tessler et al. then proposed a refined zigzag model based on Timoshenko’s beam theory [31]. The model is actually an extension of the first-order shear deformation theory, on which the piecewise linear zigzag functions are superimposed. The in-plane displacement and transverse displacement are consistent in the thickness direction, and the shear correction coefficient and shear stress continuity conditions are not required. After that, Iurlaro performed the bending and vibration behavior of the sandwich beam and plate through the first- and high- order zigzag functions [32,33]. It is found that the predicted results of the zigzag displacement model are closer to the experimental results than those of the Timoshenko’s beam model considering the shear correction factor. Furthermore, the researchers developed the refined high-order zigzag theory by means of the cubic and trigonometric functions to analyze the mechanical behaviors of laminated structures such as bending, buckling, stress, and viscoelasticity [34,35,36,37]. It is worth mentioning that the refined zigzag theory can be reduced to the classical straight line displacement field hypothesis in Allen’s works when the in-plane displacements are not considered and the rotations of the cross-section are expressed by -*λ*∂*w*/∂*x* or -*λ*∂*w*/∂*y* [38]. Allen and other related models for solving sandwich structure problems are only used to consider the sandwich structural characteristics of soft core and hard face sheets, and the assumption of the cross-section angle of face sheets and core is the same, and the shear effect of the face sheets is ignored [39].

Unlike most existing theories that ignore the shear effect of face sheets, we demonstrated in previous study [23] the necessity to consider such shear effects, especially for the case of composite face sheets under IPDL. Thus, the displacement field introduced in [23] on the basis of a zigzag hypothesis is adopted to consider the cross-section rotation angle induced by face sheet shearing, so that sufficiently accurate results can be obtained at low computational costs. Subsequently, the Hamilton principle and the Ritz method are utilized to obtain the dynamic governing equations. To compare how different cellular cores for sandwich construction perform dynamically, the geometric parameter optimizations of the metal configuration of TCOR and HHON are performed under the constraint of equal mass. Further, in the presence of IPDL, the influence of composite face sheets on fundamental frequency is explored under three different types of BCs.

## 2. Formulation

### 2.1. Problem Description

Consider the vibration of a cellular sandwich plate with a gravitational field g, as shown in Figure 1a. The origin of the Cartesian coordinate system is placed at the geometric center of the plate. Let the x-y-z directions be parallel to the width aL, height L and thickness h of the plate, respectively. Cellular foams, TCOR plates, and HHONs are successively employed as the core to construct the sandwich plate, as shown in Figure 1b. Relevant geometric parameters of the sandwich core are labeled in Figure 1b: core thickness $h_{c}$; TCOR member thickness $t_{t}$ and corrugation angle θ; HHON member thickness $t_{h}$; length $l_{h}$, and angle between horizontal line and inclined cell wall α. In the present study, only regular HHONs are discussed so that $\alpha =30^{o}$.

### 2.2. Kinematics and Constitutive Equations

Similar to our previous study [23], each type of cellular core displayed in Figure 1 is viewed as an equivalent orthogonal layer. The displacement model considering the shear deformation of face sheet layers is shown in Figure 2. Upon assuming that the points on the cross-section of the core layer and the face sheet layers have the same rotations, the blue line representing these points moves to the new position marked as the red line. The two face sheets are taken as symmetrical with identical thickness (i.e., $h_{f1}=h_{f2}=h_{f}$) and material make. Then, the rotation angles of the cross-sections in the x-z plane are denoted by $\theta_{c}$ and $\theta_{f}$ for the equivalent core and face sheet layers, respectively, as shown in Figure 2. Due to consideration of shear deformation, the rotation angles are all less than the pure bending angle dw/dx, wherein w represents the consistent transverse displacement of the equivalent core and face sheet layers.

With interface displacement continuity considered, the three constituent layers of the sandwich plate have the following displacement components in the x and y directions:a.Face sheet layer 1 $\left(-\frac{h_{c}}{2}\geq z\geq -\frac{h}{2}\right)$:



(1)
$$\begin{array}{l} u^{f1}(x,y,z)=-(z+\frac{h_{c}}{2})\theta_{f}(x,y)+\frac{h_{c}}{2}\theta_{c}(x,y)\\ v^{f1}(x,y,z)=-(z+\frac{h_{c}}{2})\lambda_{f}(x,y)+\frac{h_{c}}{2}\lambda_{c}(x,y) \end{array}$$



b.Sandwich core layer $\left(-\frac{h_{c}}{2}\leq z\leq \frac{h_{c}}{2}\right)$:



(2)
$$\begin{array}{l} u^{c}(x,y,z)=-z\theta_{c}(x,y)\\ v^{c}(x,y,z)=-z\lambda_{c}(x,y) \end{array}$$



c.Face sheet layer 2 $\left(\frac{h_{c}}{2}\leq z\leq \frac{h}{2}\right)$:

(3)$$\begin{array}{l} u^{f2}(x,y,z)=-(z-\frac{h_{c}}{2})\theta_{f}(x,y)-\frac{h_{c}}{2}\theta_{c}(x,y)\\ v^{f2}(x,y,z)=-(z-\frac{h_{c}}{2})\lambda_{f}(x,y)-\frac{h_{c}}{2}\lambda_{c}(x,y) \end{array}$$
where $\lambda_{c}$ and $\lambda_{f}$ denote the cross-section rotation angles of sandwich core layer and face sheet layers in y-z plane (Figure 2), respectively. It follows that, with small deformation assumed, the linear strain vectors of face sheets and core layer can be obtained as:(4)$$\begin{array}{cc} \varepsilon^{\left(f1,\;f2,\;c\right)} & ={\left\{\begin{array}{ccccc} \varepsilon_{x} & \varepsilon_{y} & \gamma_{xy} & \gamma_{yz} & \gamma_{xz} \end{array}\right\}}^{(f1,\;f2,\;c)\;T}\\ & ={\left\{\begin{array}{ccccc} \frac{\partial u}{\partial x} & \frac{\partial v}{\partial y} & \frac{\partial u}{\partial y}+\frac{\partial v}{\partial x} & \frac{\partial v}{\partial z}+\frac{\partial w}{\partial y} & \frac{\partial u}{\partial z}+\frac{\partial w}{\partial x} \end{array}\right\}}^{(f1,\;f2,\;c)\;T} \end{array}$$

For both the face sheets and core, the linear stress–strain relationship can be expressed as:(5)$$\begin{array}{cc} \sigma^{\left(f1,\;f2,\;c\right)} & ={\left\{\begin{array}{ccccc} \sigma_{x} & \sigma_{y} & \tau_{xy} & \tau_{yz} & \tau_{xz} \end{array}\right\}}^{(f1,\;f2,\;c)\;}{}^{T}\\ & ={\left[\begin{array}{ccccc} C_{11}^{} & C_{12}^{} & 0 & 0 & 0\\ C_{12}^{} & C_{22}^{} & 0 & 0 & 0\\ 0 & 0 & C_{44}^{} & 0 & 0\\ 0 & 0 & 0 & C_{55}^{} & 0\\ 0 & 0 & 0 & 0 & C_{66}^{} \end{array}\right]}^{\left(f1,\;f2,\;c\right)}\varepsilon^{\left(f1,\;f2,\;c\right)} \end{array}$$
where $C_{ij}^{\left(f1,\;f2,\;c\right)}$ are the elastic coefficients. As the two face sheet layers are completely symmetrical, the expressions of $C_{ij}^{f1,\;f2}$ can be expressed uniformly as: $C_{11}^{f}=\frac{E_{1}^{f}}{1-\nu_{12}^{f}\nu_{21}^{f}}$, $C_{12}^{f}=\frac{\nu_{21}^{f}E_{1}^{f}}{1-\nu_{12}^{f}\nu_{21}^{f}}$, $C_{22}^{f}=\frac{E_{2}^{f}}{1-\nu_{12}^{f}\nu_{21}^{f}}$, $C_{44}^{f}=G_{12}^{f}$, $C_{55}^{f}=G_{23}^{f}$, $C_{66}^{f}=G_{13}^{f}$. $C_{ij}^{c}$ refer to the equivalent elastic constants of sandwich core. Detailed expressions of $C_{ij}^{c}$ for TCOR plates and HHONs are given in Appendix A.

### 2.3. Energy Formulation

To investigate the vibration behavior of a cellular sandwich plate, Hamilton’s principle is adopted: $\delta \int_{t}(U+V-T)dt=0$. Therefore, the strain energy U, potential energy V, and kinetic energy T of the sandwich are given specifically as follows:Strain energy:
(6)$$\begin{array}{cc} U & =\frac{1}{2}\sum_{f1,\;f2,\;c}\int_{V}\left(\sigma_{x}\varepsilon_{x}+\sigma_{y}\varepsilon_{y}+\tau_{xy}\gamma_{xy}+\tau_{yz}\gamma_{yz}+\tau_{xz}\gamma_{xz}\right)dV\\ & =\frac{1}{2}\sum \int_{A}\int_{z}(\sigma^{f1T}\varepsilon^{f1}+\sigma^{cT}\varepsilon^{c}+\sigma^{f2T}\varepsilon^{f2})dzdA \end{array}$$

Potential energy generated from IPDLs:

(7)$$\begin{array}{cc} V_{g} & =-(\rho_{m}^{f1}g\frac{h_{f1}}{2}+\frac{\rho_{s}gh_{c}}{2}+\rho_{m}^{f2}g\frac{h_{f2}}{2})\int_{A}(L-y){\left(\frac{\partial w}{\partial y}\right)}^{2}dA\\ & =-(\rho_{m}^{f}gh_{f}+\frac{\rho_{s}gh_{c}}{2})\int_{A}(L-y){\left(\frac{\partial w}{\partial y}\right)}^{2}dA \end{array}$$
where $\rho_{m}^{f}$ is the density of the parent material of face sheet layers; $\rho_{s}$ is the equivalent density of sandwich core ($\rho_{s}^{\mathrm{HB}}$ for HHON and $\rho_{s}^{\mathrm{CR}}$ for TCOR) given by:(8)$$\begin{array}{c} \rho_{s}^{\mathrm{HB}}=\frac{2\rho_{m}^{c}t_{h}}{\sqrt{3}l_{h}}\\ \rho_{s}^{\mathrm{CR}}=\frac{\rho_{m}^{c}t_{t}\frac{2h_{c}}{\sin\theta }}{h_{c}\frac{2h_{c}}{\tan\theta }}=\frac{\rho_{m}^{c}t_{t}}{h_{c}\cos\theta } \end{array}$$
where $\rho_{m}^{c}$ is the density of the parent material of cellular sandwich core.

Kinetic energy:

(9)$$\begin{array}{ll} T & =\frac{1}{2}\sum_{f1,\;f2,\;c}\int_{V}\rho (z)\left(u_{,t}{}^{2}+v_{,t}{}^{2}+w_{,t}{}^{2}\right)dV\\ & =\frac{1}{2}\int_{A}\int_{z}(d_{,t}^{f1\;T}\rho_{m}^{f1}(z)d_{,t}^{f1}+d_{,t}^{c\;T}\rho_{s}(z)d_{,t}^{c}+d_{,t}^{f2\;T}\rho_{m}^{f1}(z)d_{,t}^{f2})dzdA \end{array}$$
where:(10)$$\begin{array}{l} d^{\left(f1,\;f2\right)}={\left\{\begin{array}{ccc} u^{f1,\;f2} & v^{f1,\;f2} & w \end{array}\right\}}^{T}\\ d^{c}={\left\{\begin{array}{ccc} u^{c} & v^{c} & w \end{array}\right\}}^{T} \end{array}$$

The subscript ‘,t’ of displacement vector $d^{\left(f1,\;f2,\;c\right)}$ represents the first derivative with respect to time t.

## 3. Solution Procedures and Validation

### 3.1. Solution Procedure

Based on the formulation of Section 2, the Ritz method is employed to determine dynamic equations of the cellular sandwich plate and the corresponding eigenvalues [23,40]. Firstly, the Ritz functions are introduced as:(11)$$\begin{array}{c} \bar{w}(\bar{x},\bar{y})=\sum_{r=0}^{p}\sum_{i=0}^{r}{\bar{w}}_{\xi }\psi_{\xi }^{w}(\bar{x},\bar{y})\sin(\omega t+\varphi )\\ \theta_{c}(\bar{x},\bar{y})=\sum_{r=0}^{p}\sum_{i=0}^{r}\theta_{\xi }^{c}\psi_{\xi }^{xc}(\bar{x},\bar{y})\sin(\omega t+\varphi )\\ \lambda_{c}(\bar{x},\bar{y})=\sum_{r=0}^{p}\sum_{i=0}^{r}\lambda_{\xi }^{c}\psi_{\xi }^{yc}(\bar{x},\bar{y})\sin(\omega t+\varphi )\\ \theta_{f}(\bar{x},\bar{y})=\sum_{r=0}^{p}\sum_{i=0}^{r}\theta_{\xi }^{f}\psi_{\xi }^{xf}(\bar{x},\bar{y})\sin(\omega t+\varphi )\\ \lambda_{f}(\bar{x},\bar{y})=\sum_{r=0}^{p}\sum_{i=0}^{r}\lambda_{\xi }^{f}\psi_{\xi }^{yf}(\bar{x},\bar{y})\sin(\omega t+\varphi ) \end{array}$$
where $\bar{w}$ is the non-dimensional transverse displacement, defined as $\bar{w}=\frac{2w}{L}$; $\bar{x}$ and $\bar{y}$ are the non-dimensional coordinates, defined as $\bar{x}=\frac{2x}{L}$, $\bar{y}=\frac{2y}{L}$; *p* is the degree of the complete polynomial space; ${\bar{w}}_{i},\;\theta_{i}^{c},\;\lambda_{i}^{c},\;\theta_{i}^{f},\;\lambda_{i}^{f}$ are unknown coefficients to be varied with subscript ξ, defined by ξ=(r+1)(r+2)/2−i; ω is the natural frequency of sandwich plate, and $\psi_{\xi }^{w},\;\psi_{\xi }^{xc},\;\psi_{\xi }^{yc},\;\psi_{\xi }^{xf},\;\psi_{\xi }^{yf}$ are the polynomial functions, which contain the basic functions $\phi^{w},\;\phi_{}^{xc},\;\phi_{}^{yc},\;\phi_{}^{xf},\;\phi_{}^{yf}$ set to satisfy the geometric BCs as:(12)$$\begin{array}{cc} \psi_{\xi }^{w,xc,yc,xf,yf} & =\phi_{}^{w,xc,yc,xf,yf}(\bar{x},\bar{y}){\bar{x}}^{i}{\bar{y}}^{r-i}\\ & ={\left(\bar{x}-a\right)}^{n1}{\left(\bar{x}+a\right)}^{n2}{\left(\bar{y}-1\right)}^{n3}{\left(\bar{y}+1\right)}^{n4}{\bar{x}}^{i}{\bar{y}}^{r-i} \end{array}$$

Here, depending upon the BC type and the direction of cross-section rotation angle, the value of nk (k=1,2,3,4) is 0 or 1.

Four different BC types for the four edges of the plate displayed in Figure 3 are involved in subsequent verification study. For example, as listed in Table 1, the boundary condition SCFF means that the 1st edge is simply supported, the 2nd edge is clamped, and the 3rd and 4th edges are free. For different combinations of BC and cross-section rotation angle direction, Table 1 lists the values of nk in order.

Secondly, upon substituting the Ritz functions (Equation (11)) into the displacement field (Equations (1)–(3)) and then applying the strain vectors (Equation (4)) and constitutive equations (Equation (5)), the energy formulas of the cellular sandwich plate can be rewritten as:(13)$$\begin{array}{l} U^{\prime }=\frac{1}{2}\sum_{f,\;c}\int_{\bar{A}}\int_{z}(\Lambda^{T}{Ν}^{(f,\;c)\;T}C^{\left(f,\;c\right)}{Ν}^{\left(f,\;c\right)}\Lambda )dzd\bar{x}d\bar{y}\\ V_{g}^{\prime }=-\frac{1}{2}\int_{\bar{A}}(1-\bar{y})\Lambda^{T}{{Ν}^{\prime }}^{T}\bar{q}{Ν}^{\prime }\Lambda d\bar{x}d\bar{y}\\ T^{\prime }=\frac{1}{2}\int_{\bar{A}}\int_{z}\left(\Lambda_{,t}^{T}D^{(f,\;c)\;T}\rho (z)D^{\left(f,\;c\right)}\Lambda_{,t}\right)dzd\bar{x}d\bar{y} \end{array}$$
where ${Ν}^{\left(f,\;c\right)}$, ${Ν}^{\prime }$, $D^{\left(f,\;c\right)}$ are the non-dimensional coefficient matrices of strain vectors and displacement vectors. These matrices can be conveniently obtained from Equations (1)–(4) and (11), thus their specific expressions are not presented herein for brevity. Λ is the vector of unknown coefficients, expressed as ${\left[\begin{array}{ccccc} \theta_{\xi }^{c} & \theta_{\xi }^{f} & \lambda_{\xi }^{c} & \lambda_{\xi }^{f} & {\bar{w}}_{\xi } \end{array}\right]}^{T}\sin(\omega t+\varphi )$. $\bar{q}$ is the non-dimensional parameter quantifying IPDLs, defined as $\frac{\rho gL^{3}}{D_{\mathrm{eq}}}$; ρ represents the mass per area of sandwich plate in the x-y plane, and $D_{\mathrm{eq}}$ is the parameter quantifying the ability of the sandwich plate to resist bending in the y-direction, given by:(14)$$\begin{array}{c} \rho =\int_{-h_{c}/2}^{h_{c}/2}\rho_{s}dz+2\int_{h_{c}/2}^{h/2}\rho_{m}^{f}dz\\ D_{\mathrm{eq}}=\int_{-h_{c}/2}^{h_{c}/2}z^{2}C_{22}^{c}dz+2\int_{h_{c}/2}^{h/2}z^{2}C_{22}^{f}dz \end{array}$$

Finally, by substituting the new energy formulas of Equation (13) into Hamilton’s principle and performing variational calculations on unknown coefficients Λ, the dynamic equations of the cellular sandwich plate expressed in the form of an eigenvalue problem are obtained as:(15)$$(K+\bar{q}K_{g}-{\bar{\omega }}^{2}M)\Lambda =0$$
where $\bar{\omega }$ is the non-dimensional natural frequency, defined as $\sqrt{\frac{\rho \omega^{2}L^{4}}{D_{\mathrm{eq}}}}$. The structural stiffness matrix K, the geometric stiffness matrix $K_{g}$ induced by IPDLs, and the mass matrix M are all 5th order symmetric matrices, while detailed expressions of K and M are presented in Appendix B and Appendix C, respectively. The geometric stiffness matrix is given by:(16)$$K_{g}=-\int_{\bar{A}}(1-\bar{y}){B^{\prime }}^{T}B^{\prime }d\bar{x}d\bar{y}$$

In the current study, only changes in the first-order natural frequency and modal shape of a cellular sandwich plate are concerned, which can be obtained by solving the eigenvalue problem of Equation (15). With the mass matrix omitted, the critical buckling IPDL can be obtained by calculating the minimum eigenvalue of $\left(K,\;K_{g}\right)$.

### 3.2. Validation

In this section, the three-dimensional (3D) theory of free vibration established above for cellular-cored sandwich plates subjected to linearly varying in-plane distributed loads are validated.

Consider first the limiting case of sandwich plates, that is, the monolithic plate. For a monolithic rectangular plate (a = 0.5; Figure 1), Table 2 compares its critical buckling IPDL $\bar{q}$ and natural fundamental frequency $\bar{\omega }$ calculated from the present method with existing theoretical predictions by Wang et al. [13,14] and Yu et al. [16,17]. Additionally included in Table 2 are 3D finite element (FE) simulation results obtained using the commercially available FE code ABAQUS. As shown in Table 2, overall, the present results agree well with existing theoretical predictions and FE calculations, irrespective of the BCs. Nonetheless, the values obtained for both $\bar{q}$ and $\bar{\omega }$ using the proposed theory are consistently smaller than those of existing theoretical predictions that did not consider shear effect of the monolithic plate, thus more consistent with 3D FE simulation results. Further, while the present predictions are sensitive to the slenderness ratio L/h of the monolithic plate, such sensitivity is absent in the theoretical predictions of Wang et al. [13,14] and Yu et al. [16,17].

To further validate the proposed theory, direct FE simulations of cellular sandwich plates are performed, with 3D deformable four-node shell elements and eight-node solid elements (S4R, C3D8R) selected to model the sandwich core and the face sheets, respectively. Interactions of Tie are exerted between the face sheets and core layer. For simply supported sandwich plates, the freedoms of reference point located at the geometric center of the BC face are constrained, while the points in the BC face are coupled with the reference point. For sandwich plates having either clamped or free edges, the BCs can be straightforwardly implemented in ABAQUS.

For each FE simulation, a linear frequency analysis step is used to extract the natural frequencies and modal shapes. To consider the influence of IPDL, a prior static mechanical analysis before Frequency step is performed. At the static analysis step, a gravity load is applied and the option of large deformation is turned on. For calculating the critical buckling IPDL, only a linear buckle analysis step is required [10,23]. For both face sheets and core, aluminum alloy is selected as the material make, with density $\rho_{\mathrm{Al}}=2700{\;\mathrm{kg}/m}^{3}$, Poisson ratio ν=0.3, and Young’s modulus $E_{\mathrm{Al}}=70\;\mathrm{GPa}$. Irrespective of core type, the sandwich plate has the following geometrical parameters: L=4 m, $h_{c}=0.09\;m$, h=0.1 m, and a=1. It follows that the sandwich core has a relative density of ${\bar{\rho }}_{s}=\rho_{s}/\rho_{m}=0.058$.

Figure 4 compares the theoretical and numerical results obtained for both TCOR and HHON sandwich plates in terms of $\bar{\omega }$ versus $\bar{q}$ curves with excellent agreement achieved, irrespective of BCs considered, SSSS, FSFS, and SCFF. With the value of $\bar{q}$ fixed, the sandwich plate with SSSS exhibits the largest $\bar{\omega }$, followed in order by those with FSFS and SCFF.

## 4. Results and Discussion

The proposed 3D vibration theory, validated in the previous section, is applied in this section to investigate how the BC and key geometrical/material parameters (e.g., core thickness $h_{c}$, core density $\bar{\rho_{s}}$, corrugation angle θ, and face sheet material) of a cellular sandwich plate affects vibration behaviors. Firstly, how varying the BC of the sandwich plate affects its fundamental frequency under IPDL is quantified. Secondly, how the fundamental frequency depends upon geometric parameters is evaluated, with equal mass assumed. Thirdly, the frequency versus IPDL curves for four different kinds of plates under varying BCs are compared: monolithic plate, foam sandwich plate, TCOR sandwich plate, and HHON sandwich plate. Further, the material make of the face sheets is varied from metal to fiber-reinforced composite to see how this would affect its vibration performance. Lastly, first-order vibration modal shapes of sandwich plates are discussed.

### 4.1. Effect of Boundary Condition (BC)

For TCOR and HHON sandwich plates, the results of Figure 4 reveal that different types of BCs have similar influence on plate vibration under IPDL. With the increase of dimensionless critical buckling IPDL $\bar{q}$, the dimensionless natural fundamental frequency $\bar{\omega }$ exhibits a different downward variation trend as the BC is varied. A sandwich plate with SSSS BC is the most stable and possesses the highest frequency as most degrees of freedom of its four edges are constrained. As for the sandwich plate with two free edges (i.e., either FSFS or SCFF BC), the plate is more stable with opposite edges constrained (FSFS), even though the clamped bottom edge in the constrained adjacent edges (SCFF) restricts more degrees of freedom than the simply supported edge in FSFS. However, as $\bar{q}$ is increased, $\bar{\omega }$ drops faster under FSFS than that under SCFF because the simply supported edge in SCFF is parallel to the direction of IPDL.

### 4.2. Effect of Geometric Parameters

To compare the mechanical properties (i.e., fundamental frequency and critical buckling IPDL) between TCOR and HHON sandwich plates having equal mass, geometric parameter optimization is performed first. With the cost of computation in mind, only SSSS and FSFS BCs are considered in the optimization. The face sheet thickness is fixed at $L/2h_{f}=400$. To visualize the discrepancy among the calculated curves more clearly, non-dimensional natural fundamental frequency $100\omega L\sqrt{{\left(\rho /E\right)}_{\mathrm{Al}}}$ and non-dimensional IPDL $1000gL{\left(\rho /E\right)}_{\mathrm{Al}}$ are introduced, with the subscript ‘Al’ denoting aluminum alloy. As the mass of face sheets is fixed, using the expressions of equivalent density of sandwich core as shown in Equation (8) leads to the non-dimensional mass of HHON core ${\bar{M}}_{\mathrm{HB}}$ and that of TCOR core ${\bar{M}}_{\mathrm{CR}}$ as:(17)$${\bar{M}}_{\mathrm{HB}}=\frac{M_{\mathrm{HB}}}{M_{f}}=\frac{\rho_{s}^{\mathrm{HB}}h_{c}L^{2}}{\rho_{m}h_{f}L^{2}}={\bar{\rho }}_{s}^{\mathrm{HB}}\frac{h_{c}}{h_{f}}=\frac{2t_{h}}{\sqrt{3}l_{h}}\frac{h_{c}}{h_{f}}$$
(18)$${\bar{M}}_{\mathrm{CR}}=\frac{M_{\mathrm{CR}}}{M_{f}}=\frac{\rho_{s}^{\mathrm{CR}}h_{c}L^{2}}{\rho_{m}h_{f}L^{2}}={\bar{\rho }}_{s}^{\mathrm{CR}}\frac{h_{c}}{h_{f}}=\frac{t_{t}}{h_{c}\cos\theta }\frac{h_{c}}{h_{f}}$$
where $M_{\mathrm{HB}}$ and $M_{\mathrm{CR}}$ represent separately the mass of HHON core and TCOR core, and $M_{f}$ represents the mass of a single face sheet. Equation (17) implies that the geometry and mass of HHON core with specified ${\bar{M}}_{\mathrm{HB}}$ can be determined by two independent ratios, that is, ${\bar{\rho }}_{s}^{\mathrm{HB}}$ and $h_{c}/h_{f}$. The variations of ${\bar{\rho }}_{s}^{\mathrm{HB}}$ correspond to changes in $t_{h}/l_{h}$. The thickness $t_{h}$ or length $l_{h}$ of the honeycomb cell wall can be determined as one of that which is given. $h_{c}/h_{f}$ determines the thickness of the sandwich core.

Unlike the HHON core, two independent ratios, that is, ${\bar{\rho }}_{s}^{\mathrm{CR}}$ and $h_{c}/h_{f}$, determine the mass of a TCOR core, but not its geometry, because ${\bar{\rho }}_{s}^{\mathrm{CR}}$ contains two separate variables: $t_{t}/h_{c}$ and θ. Hence, for the TCOR core, the influence of $t_{t}/h_{c}$ and θ on the frequency versus IPDL curve is firstly discussed under the constraint of equal mass, as shown in Figure 5, with ${\bar{M}}_{\mathrm{CR}}=1.04$, ${\bar{\rho }}_{s}^{\mathrm{CR}}=0.058$, and $h_{c}/h_{f}=18$. For both the SSSS and FSFS BCs, representative angles of π/6, π/4 and π/3 are discussed. The results of Figure 5 show that the TCOR sandwich plate exhibits slightly higher frequencies at the combination of bigger inclination angle and thinner corrugation member thickness. However, a larger corrugation angle makes it more difficult to fabricate the sandwich. Hence, in the rest of this study, the corrugation angle is set to π/3 in consideration of processing.

Figure 6 compares the frequency versus IPDL curves of TCOR and HHON sandwich plates with equal mass (${\bar{M}}_{\mathrm{CR}}={\bar{M}}_{\mathrm{HB}}=1.04$) at different combinations of core density ${\bar{\rho }}_{s}$ and thickness $h_{c}/h_{f}$. Both the fundamental frequency and critical buckling IPDL decrease as core equivalent density is increased (or, equivalently, core thickness is decreased). To a certain extent, as the sandwich plate will degenerate to a monolithic plate as its core density is increased or core thickness is decreased, these results suggest the superiority of sandwich plate over its monolithic counterpart.

The results of Figure 6 reveal that both the critical buckling IPDL and fundamental frequency of TCOR sandwich plates are higher than those of HHON sandwich plates under either SSSS or FSFS BC. The discrepancy between the two sandwich types becomes negligible as core density is increased or core thickness is decreased, since both TCOR and HHON sandwich plates approach a monolithic plate in the limit. Moreover, the superiority of TCOR core to HHON core is more obvious under FSFS BC, compared with SSSS BC, for the TCOR core possesses stronger elastic constants in the vertical direction, resulting in its much higher structural stiffness of the sandwich plate under FSFS BC.

Essentially, changes in the curves of Figure 5 and Figure 6 are attributed to variations of elastic constants when the core type of cellular sandwich plate is varied. As compared in Figure 7, the great in-plane elastic anisotropy exhibited by a TCOR core can be seen intuitively from the discrepancy between $C_{22}^{c}$ and $C_{11}^{c}$ (or $C_{12}^{c}$), the former over ten times larger than the latter. Under either FSFS or SSSS BC, this great anisotropy enables a TCOR sandwich plate to become superior relative to an HHON sandwich. Moreover, the in-plane shear elastic constant $C_{44}^{c}$ and the transverse shear constant $C_{55}^{c}$ of the HHON core are much smaller than those of the TOCR core. This also contributes to the relatively poor performance of an HHON sandwich plate. Overall, the better stability of a TCOR sandwich plate is be attributed to its higher elastic constants, that is, $C_{22}^{c}$, $C_{44}^{c}$, and $C_{55}^{c}$.

### 4.3. Comparison between Metal and Composite Face Sheets

In nature and engineering applications, HHON sandwiches are more commonly found than TCOR sandwiches. Therefore, in this section, for enhanced in-plane mechanical performance, the metallic face sheets are replaced by composite face sheets to construct HHON sandwich plates. The composite face sheets are made of carbon fiber-reinforced composite T700/3234, with its mechanical properties given by Ref. [41]: $E_{1}=110\;\mathrm{GPa}$, $E_{2}=E_{3}=8.7\;\mathrm{GPa}$, $G_{12}=G_{13}=G_{23}=4\;\mathrm{GPa}$, $\nu_{13}=0.3$, and $\nu_{21}=\nu_{23}=0.32$. Figure 8 illustrates three different fiber stacking types of composite face sheets for HHON sandwich construction.

As shown in Figure 9a for SSSS BC, under the constraint of equal mass, an HHON sandwich plate with composite face sheets has lower fundamental frequency than its counterpart with Al face sheets. Further, varying fiber stacking changes the anisotropy of the HHON sandwich plate, thus also changing its fundamental frequency. Simultaneously, the critical buckling IPDL is dependent upon the degree of in-plane anisotropy. In particular, when the fiber direction is parallel to the IPDL, that is, the Y-direction fiber, the sandwich plate is more stable.

In contrast, under FSFS BC as shown in Figure 9b, both the frequency and critical buckling IPDL are higher for a sandwich plate having Y- or XY-direction fibers laying in composite face sheets, compared to its counterpart with Al face sheets. This is because under the FSFS BC, the advantage of a stronger stiffness (i.e., Y-direction fiber stacking) in the y-direction can be fully exploited.

Figure 9c compares the frequency versus IPDL curves of different sandwich plates with equal mass under SFSF BC, including monolithic plate, all-metallic cellular sandwich plates with different cores, and cellular sandwich plates with composite face sheets. For all the plates considered here, the fundamental frequency remains nearly constant as the IPDL is increased (called Stage I), and then decreases when the IPDL exceeds a specific value (called Stage II). (This transition from Stage I to Stage II may be attributed to the shift of modal shape as the IPDL is increased under SFSF BC, as discussed in Section 4.5.) When the dimensionless IPDL, $g/g_{HOM}$, is less than 10, the dimensionless frequency of the HHON sandwich plate, $\omega /\omega_{HOM}$, is almost the same as that of the TCOR sandwich. As the $g/g_{HOM}$ exceeds 10, the $\omega /\omega_{HOM}$ of the TCOR sandwich becomes larger than that of the HHON sandwich. However, upon enhancing the x-direction structural stiffness via X-direction and XY-direction fiber stacking in composite face sheets, the fundamental frequency of the HHON sandwich plate remarkably increases, although its critical buckling IPDL decreases.

It may be concluded from the results of Figure 9 that increased frequency and critical buckling IPDL can be realized only under partial BCs for sandwich plates with composite face sheets that have specific fiber stacking types.

Next, to demonstrate the necessity of considering the shear effect of face sheets for both vibration and buckling analysis, Figure 10 compares the results between TCOR sandwich plates with and without considering such shear effect. When shear effect is ignored, the cross-section rotation angles $\theta_{f}$ and $\lambda_{f}$ in Equations (1) and (3) representing the shear effect of face sheets are replaced with ∂w/∂x and ∂w/∂y. From Figure 10a, it can be seen that the difference of frequency between cases with and without considering shear effect is small when the face sheets are thin, even less than 2%. However, the difference increases significantly as the face sheets become thicker, which means that shear effect becomes important for relatively thick face sheets. At the same time, the critical buckling IPDLs of Figure 10b reveal that the difference between the two groups of results is about twice as large as that in frequency. This means that the shear effect of face sheets becomes more important when the IPDL is considered.

### 4.4. Effect of Sandwich Core Type

Figure 9 compares the frequencies and critical buckling IPDLs among the four different types of plate having equal mass: homogeneous (monolithic) plate, aluminum foam-cored sandwich plate, TCOR-cored sandwich plate, and HHON-cored sandwich plate. For simplicity, the face sheets of all the plates are made of Al alloy. The aluminum foam has closed cells and the following material properties: $\rho_{\mathrm{Al}\text{-}f}=540{\;\mathrm{kg}/m}^{3}$, $\upsilon_{\mathrm{Al}\text{-}f}=0.3$, and $E_{\mathrm{Al}\text{-}f}=405\;\mathrm{MPa}$. From Figure 9, it is seen that the frequency of foam-cored sandwich plate is almost thrice than that of the homogeneous plate. As for the TCOR and HHON sandwich plates, the deviation is even larger: the frequencies are more than four times that of the homogeneous plate. Further, the sandwich plates have much larger critical buckling IPDLs than the homogeneous plate, confirming that the former is significantly more stable than the latter. Among the three cellular core types, the TCOR and HHON are more efficient in enhancing the structural stability of a sandwich construction than the foam. In addition, with the highest frequency and largest critical buckling IPDL, the TCOR sandwich plate outperforms the others.

Comparing the structural characteristics of the sandwich plates with different core types, the TCOR and HHON sandwich cores both have the higher thickness than the foam core under the equal mass constraint, since the equivalent core density of either TCOR or HHON is much smaller than that of foam. Therefore, the increase of the core thickness is a benefit to the improvement of the stability of the sandwich plate because it can increase the overall flexural stiffness of the sandwich structure. The flexural stiffness is proportional to the cubic thickness of the structure. However, the premise is that when the loads act, the core plate of TCOR and HHON cannot exhibit local buckling. In other words, the core should not be too thin. Moreover, if the core thickness is kept constant and only the core density is increased, the critical buckling IPDLs of the structure will increase, but the fundamental frequency will first increase and then decrease. As the core density increases the overall stiffness of the structure, the mass of the structure also increases. A similar structural parametric study can be referred to in our previous study [37,42].

### 4.5. Comparison of Modal Shapes

Figure 11 and Figure 12 present the FE simulated first-order vibration modal shapes of TCOR and HHON sandwich plates subjected to IPDL under different BCs. Overall, the presence of IPDL shifts the maximum modal displacement downward from the plate center. Especially for the cases under SFSF BC, the presence of IPDL leads to an obvious change in modal shape.

Figure 13 displays the theoretically predicted contours of first-order vibration modal shapes considering IPDLs, which are consistent with the FE simulation results of Figure 11 and Figure 12. This again confirms the viability of the three-dimensional vibration theory developed in the present study for cellular-cored sandwich plates. Specifically, as the IPDLs increased, the modal shift of a sandwich plate under SFSF is presented in Figure 14. It is seen that one half-wave along the y-direction evolves into two half-waves.

As previously mentioned, this modal shift corresponds to the transition of the curves displayed in Figure 9c. At the same time, this phenomenon did not occur under SSSS and FSFS BCs when the IPDLs increased. This is because the out-plane displacements of the top and bottom edge are simultaneously constrained, and a wave parallel to the direction of the IPDLs exists in their modal shapes. This will cause their fundamental frequency to drop faster than the SFSF BC under IPDLs. The presence of a wave perpendicular to the direction of the IPDLs can reduce the fundamental frequency affected by the IPDLs. The results in Figure 4 and Figure 9 can prove this phenomenon as the fundamental frequency under SSSS BC drops slower than FSFS BC, and that under SFSF BC is the slowest when the IPDLs are not particularly strong. Moreover, the maximum modal displacement of the modal shapes all move downward from the plate center under the considered BCs. However, the constraints of the out-plane displacement on the top or bottom edge constrained the infinite deformation (an ideal situation) when IPDLs keep increasing. However, the ‘infinite deformation’ can occur under SFSF BC when the IPDLs act on the whole plate area and keep increasing, which looks like a plate rotates around the *x* axis. Then, an ‘S’ wave paralleled to the direction of IPDLs appeared, and the fundamental frequency drops faster as shown in Figure 9c.

## 5. Conclusions

With cross-section rotation angles of both the face sheets and core layer considered, a three-dimensional (3D) theory is established to characterize the vibration performance of ultralight cellular-cored sandwich plates subjected to linearly varying in-plane distributed loads (IPDLs) under different boundary conditions (BCs). For validation, 3D finite element simulations are carried out, with good agreement between theory and simulation achieved. The influence of sandwich core type on the fundamental frequency of the sandwich plate is quantified, including close-celled foams, triangular corrugated (TCOR) metal plates, and metal hexagonal honeycombs (HHON). Systematic parametric study is conducted to explore the influence of the key geometric parameters (e.g., core thickness $h_{c}$, core density $\bar{\rho_{s}}$, and corrugation angle θ) of the metal cellular sandwich core, the mixture application of composite face sheets, and the BCs on fundamental frequency and critical buckling IPDL. The main conclusions are summarized as follows.

(1)The fundamental frequency and critical buckling IPDL of a sandwich plate are both much higher than a monolithic plate with equal mass. As the sandwich core, TCOR and HHON are more efficient in enhancing the structural stability than the foam.(2)For TCOR metal sandwich plates, the frequency and critical buckling IPDL are not sensitive to the inclination angle of corrugation. However, the frequency and critical buckling IPDL of both TCOR and HHON sandwich plates are quite sensitive to either core density or core thickness.(3)The frequency versus IPDL curves and the vibration modal shapes are quite different for sandwich plates under different BCs (i.e., SSSS, SFSF, and FSFS). Especially for the case under SFSF BC, the vibration modal may shift as the IPDL is increased.(4)Using fiber-reinforced composite face sheets in lieu of metal face sheets enhances the performance of the HHON sandwich plate in terms of fundamental frequency and critical buckling IPDL. However, such enhancement occurs only under partial BCs with specific types of fiber stacking.

## Figures and Tables

**Figure 1 materials-16-04086-f001:**
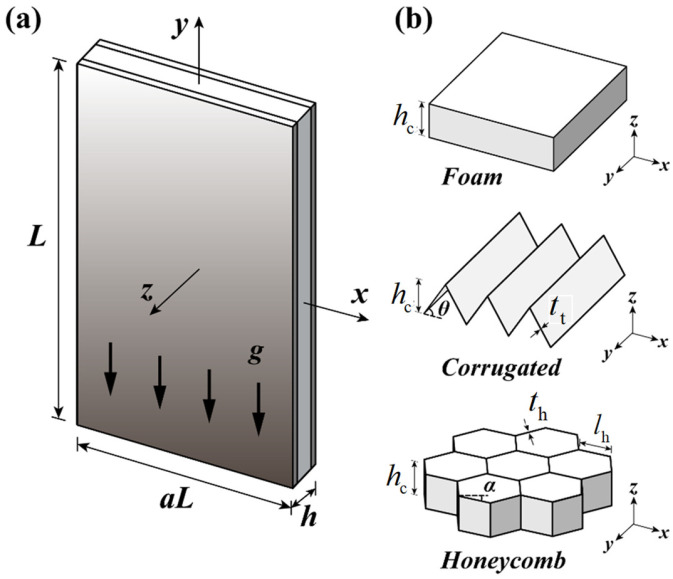
(**a**) Cellular sandwich plate subject to linearly varying in-plane distributed load (IPDL) and (**b**) cellular foam, triangular corrugated (TCOR) metal plate, and hexagonal metal honeycomb (HHON) as sandwich core.

**Figure 2 materials-16-04086-f002:**
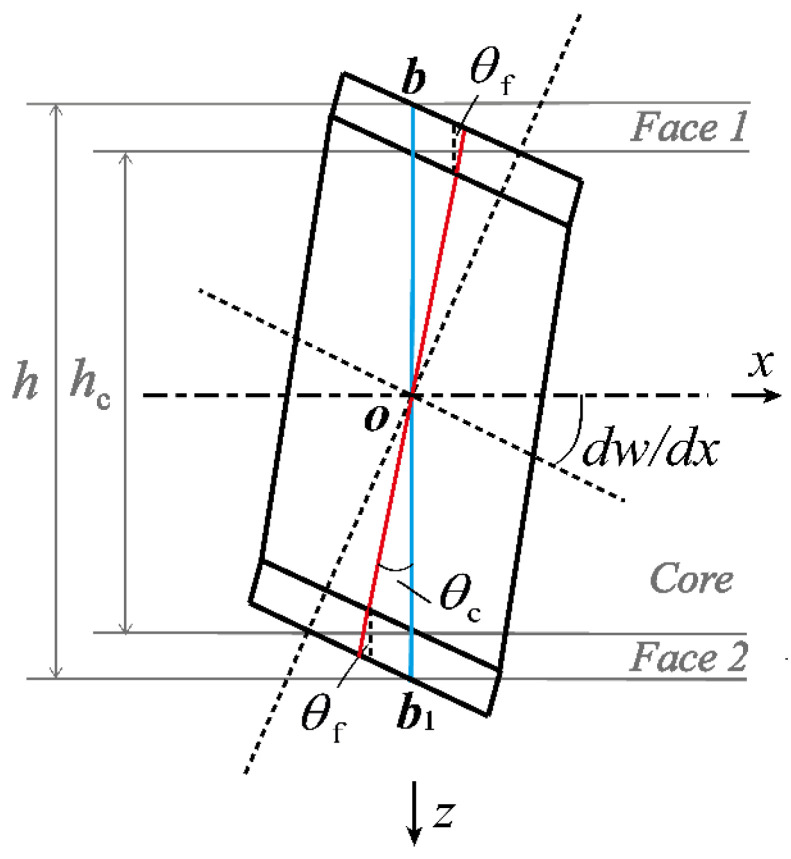
Displacement model considering cross-sectional rotation due to shear of face sheet layers (x-z plane) [23].

**Figure 3 materials-16-04086-f003:**
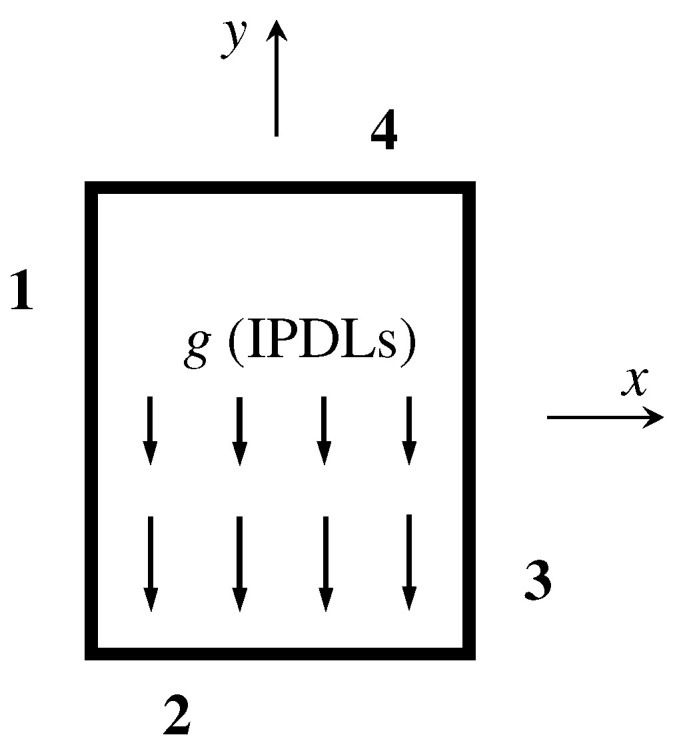
Order of plate edges for boundary conditions specified in Table 1.

**Figure 4 materials-16-04086-f004:**
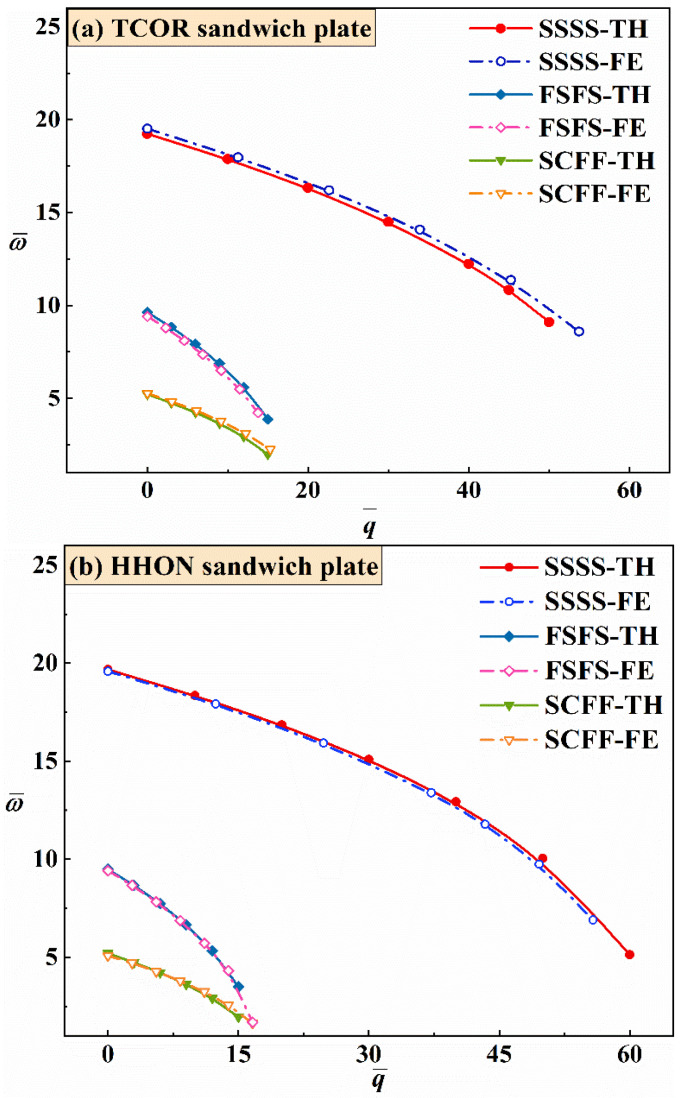
Comparison between theoretical predictions (TH) and FE simulations (FE): natural fundamental frequency $\bar{\omega }$ plotted as a function of critical buckling IPDL parameter $\bar{q}$ for (**a**) TCOR and (**b**) HHON sandwich plates under selected BCs.

**Figure 5 materials-16-04086-f005:**
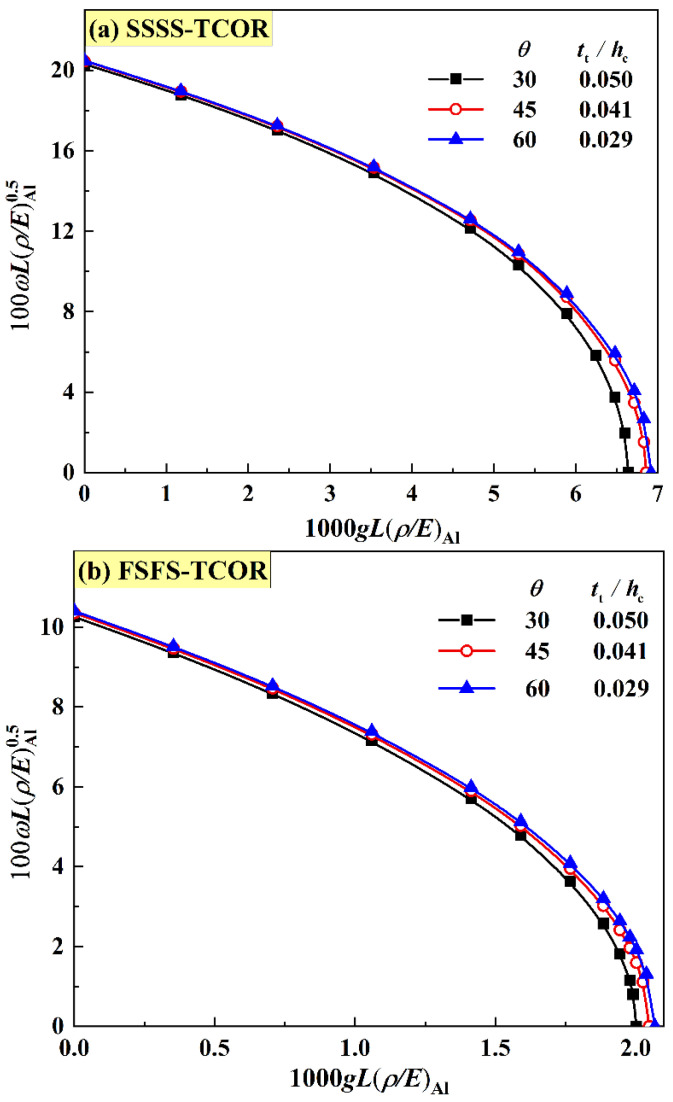
Effect of inclination angle on frequency versus IPDL curves for TCOR sandwich plates having equal mass (${\bar{M}}_{\mathrm{CR}}=1.04$, ${\bar{\rho }}_{s}^{\mathrm{CR}}=0.058$, $h_{c}/h_{f}=18$ ) with (**a**) SSSS BC and (**b**) FSFS BC.

**Figure 6 materials-16-04086-f006:**
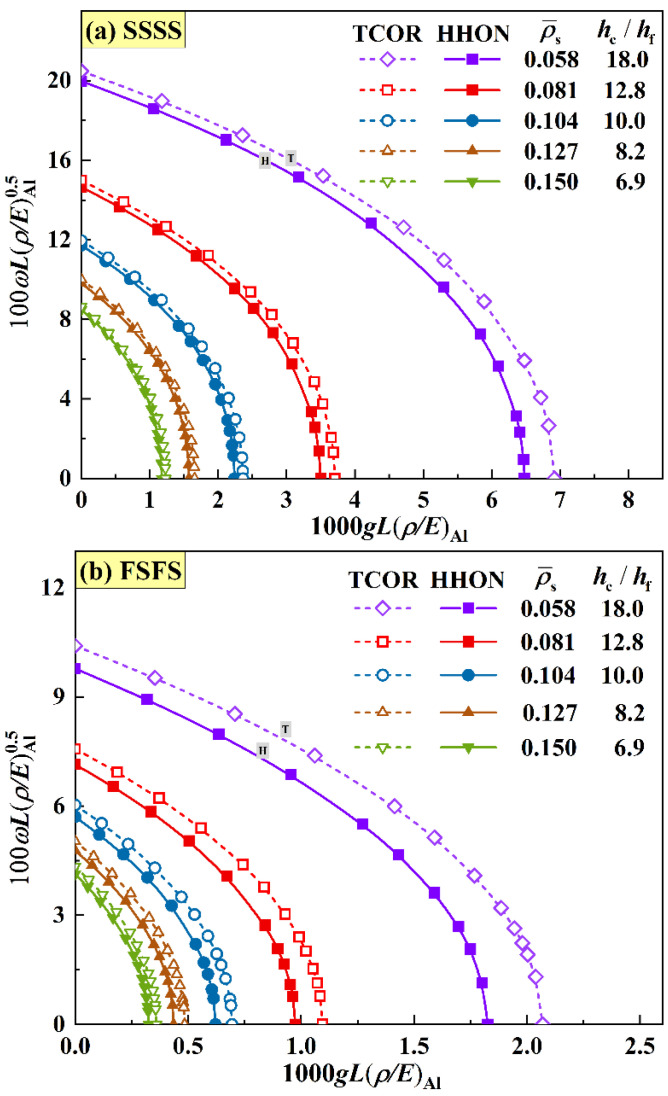
Effect of core thickness $h_{c}/h_{f}$ or equivalent core density ${\bar{\rho }}_{s}$ on frequency versus IPDL curves of TCOR and HHON sandwich plates under equal mass (${\bar{M}}_{\mathrm{CR}}={\bar{M}}_{\mathrm{HB}}=1.04$, θ=π/3 ) with (**a**) SSSS BC and (**b**) FSFS BC.

**Figure 7 materials-16-04086-f007:**
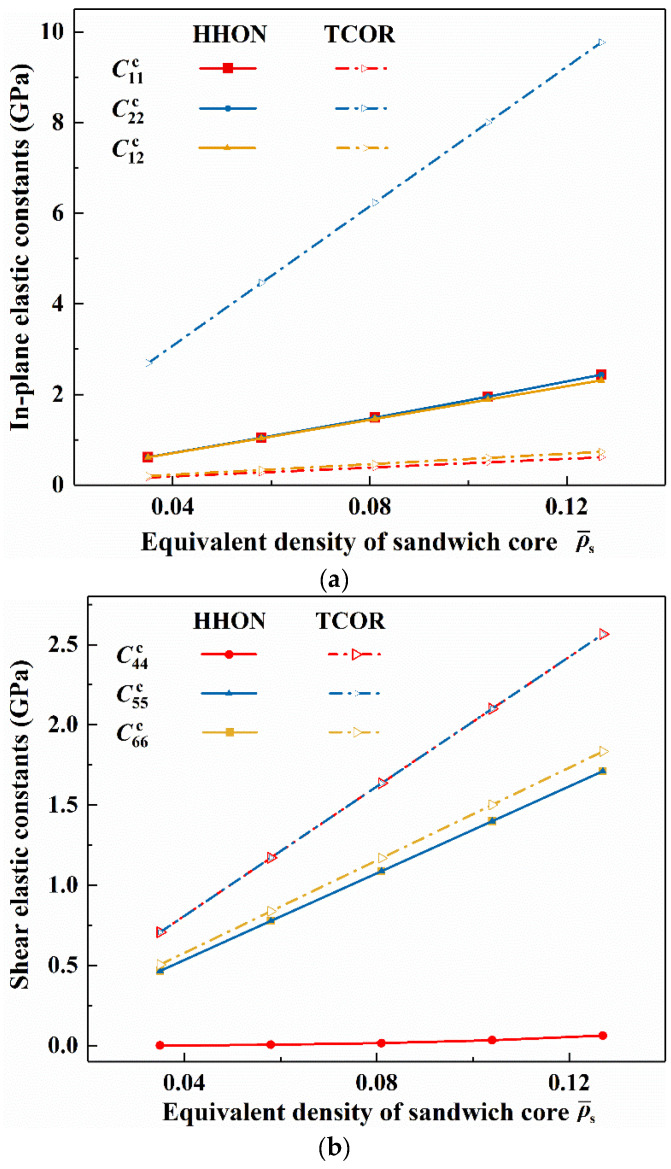
(**a**) In-plane and (**b**) transverse elastic constants plotted as functions of equivalent density: comparison between TCOR and HHON cores.

**Figure 8 materials-16-04086-f008:**
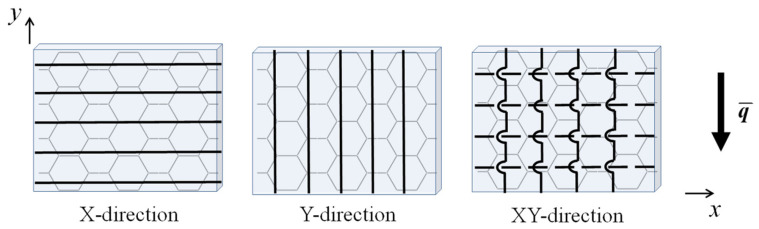
Fiber stacking types of composite face sheets for sandwich plate construction: X-direction and Y-direction refer to a single layer of fiber laying in x- and y-direction, respectively; XY-direction denotes two stacking layers of fibers laying in x- and y-direction, respectively, with x-direction fibers placed inside and closer to sandwich core.

**Figure 9 materials-16-04086-f009:**
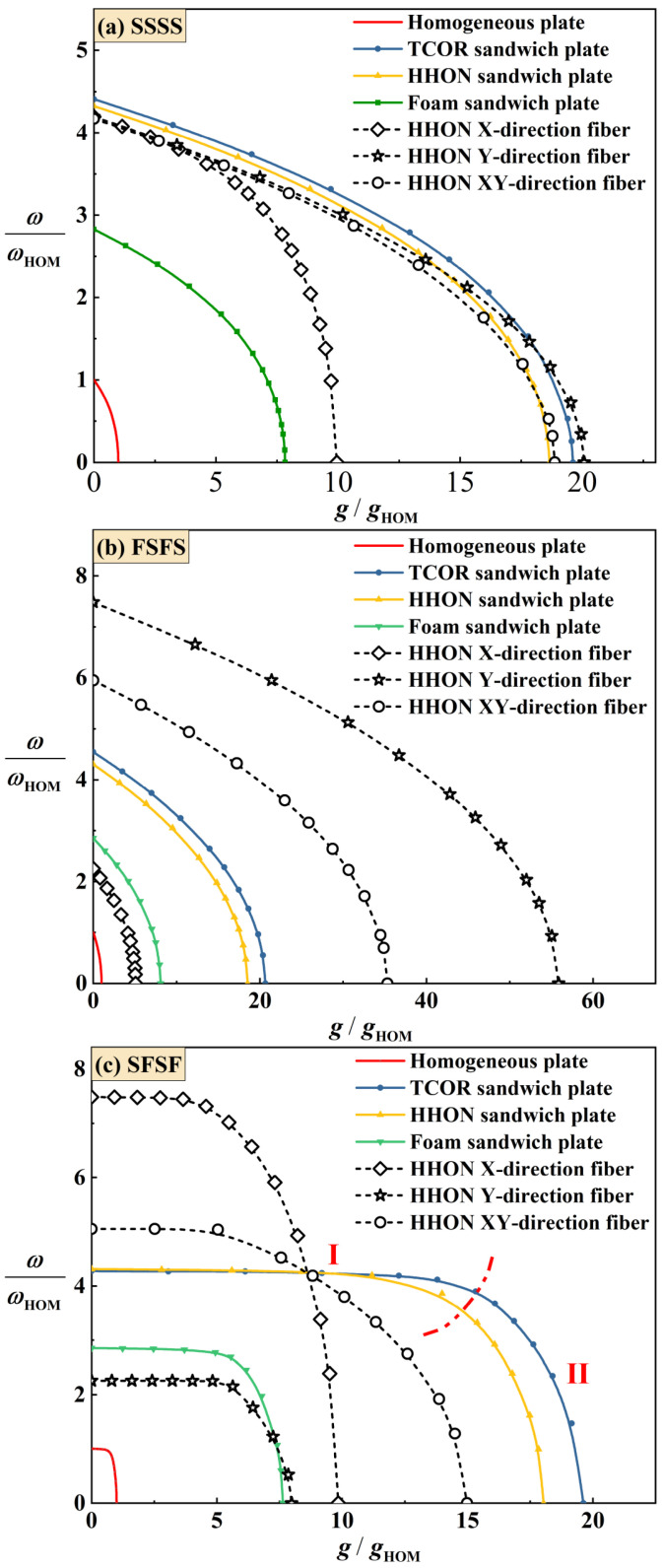
Comparison of dimensionless frequencies of different sandwich plates having equal mass (with ${\bar{M}}_{\mathrm{CR}}=1.04$, ${\bar{\rho }}_{s}^{}=0.127$, $h_{c}/h_{f}=8.2$ ). $\omega_{\mathrm{HOM}}$ and $g_{\mathrm{HOM}}$ denote the fundamental frequency and critical buckling IPDL of a homogeneous plate, respectively. The face sheets are made of either Al alloy (solid lines) or fiber-reinforced composite laminates (T700/3234, dotted lines).

**Figure 10 materials-16-04086-f010:**
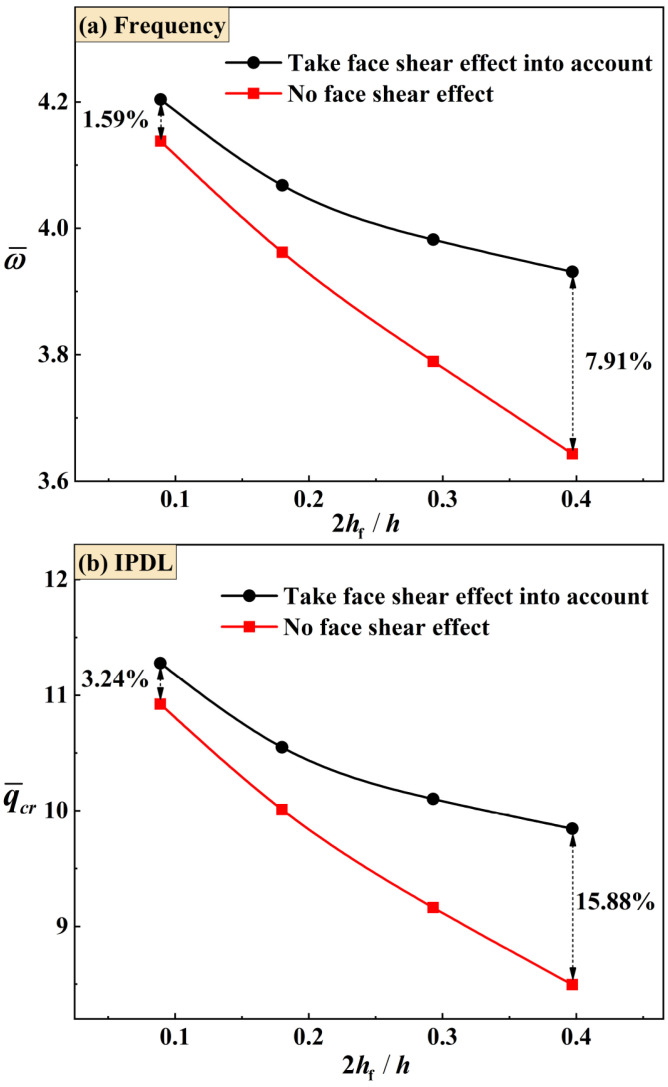
The calculated (**a**) natural frequency and (**b**) critical buckling IPDL for TCOR sandwich plates under SCFF BC, considering both the case with the shear effect of face sheets and that without the shear effect of face sheets. The face sheets are made by the XY-direction fiber composite laminates.

**Figure 11 materials-16-04086-f011:**
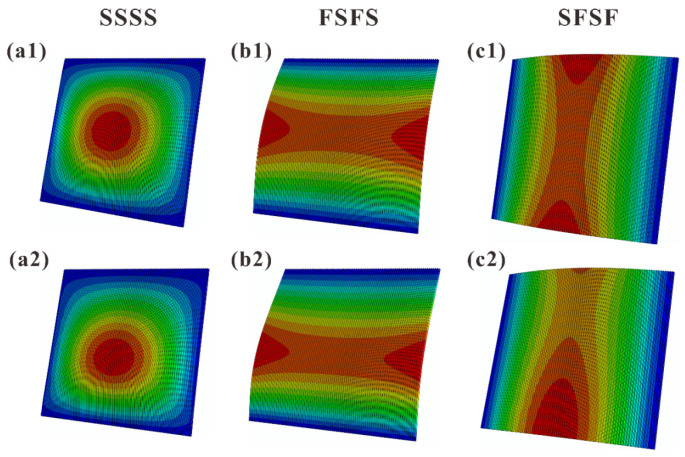
FE calculated first-order vibration modal shapes of TCOR sandwich plates under three types of BC: (**a1**–**c1**) IPDL not considered; (**a2**–**c2**) IPDL considered.

**Figure 12 materials-16-04086-f012:**
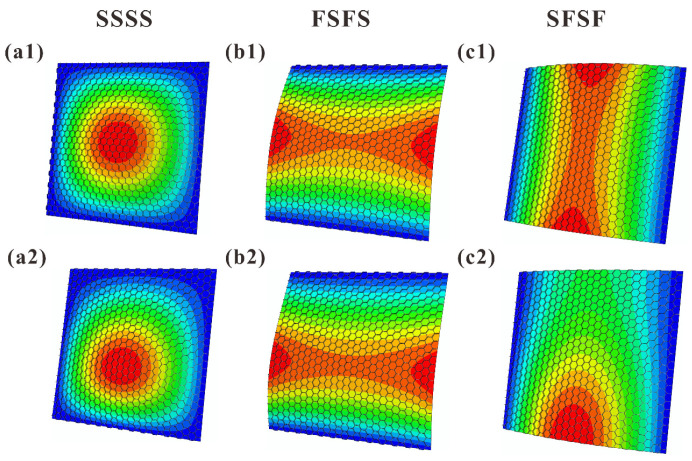
FE calculated first-order vibration modal shapes of HHON sandwich plates under three types of BC: (**a1**–**c1**) IPDL not considered; (**a2**–**c2**) IPDL considered.

**Figure 13 materials-16-04086-f013:**
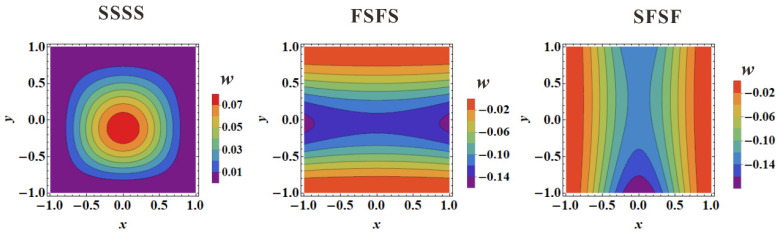
Theoretically predicted contours of first-order vibration modal shapes for sandwich plates subject to IPDL under three types of BC.

**Figure 14 materials-16-04086-f014:**
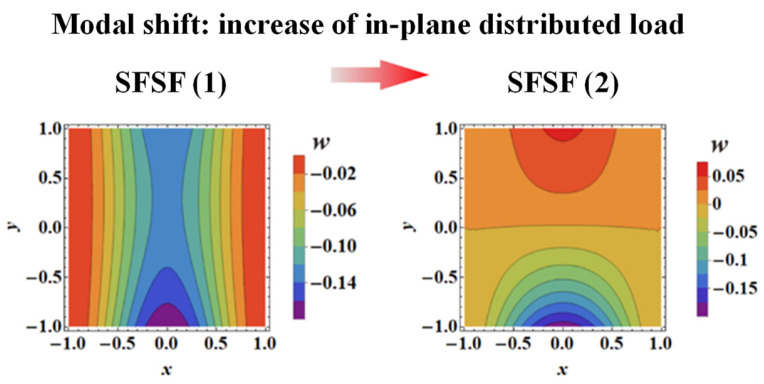
First-order vibration modal shape of a sandwich plate with SFSF BC subjected to: (1) small IPDL and (2) large IPDL.

**Table 1 materials-16-04086-t001:** Values of nk (k=1,2,3,4) for selected combinations of cross-section rotation angle direction and BC.

BC	$$\phi_{}^{xc},\phi_{}^{xf}$$	$$\phi_{}^{yc},\phi_{}^{yf}$$	$$\phi^{w}$$
SSSS (1234)	0,0,1,1	1,1,0,0	1,1,1,1
FSFS (1234)	0,0,1,1	0,0,0,0	0,0,1,1
SFSF (1234)	0,0,0,0	1,1,0,0	1,1,0,0
SCFF ^a^ (1234 ^b^)	0,0,0,1	0,1,0,1	0,1,0,1

^a^ S—simply supported (rotational degree of freedom around the edge-axis unconstrained); C— clamped (six degrees of freedom constrained); F—free (no degree of freedom constrained). ^b^ Order of plate edges are specified in Figure 3.

**Table 2 materials-16-04086-t002:** Dimensionless natural fundamental frequency ($\bar{\omega }$) and critical buckling IPDL ($\bar{q}$) of a monolithic plate (a = 0.5): comparison between the present predictions and existing theoretical and FE results.

Case	Method	SSSF	SFSF	SCSF	SSSS
$\begin{array}{c} \bar{\omega }\\ \left(\bar{q}=0\right) \end{array}$	FE* (L/h = 40)	40.208	38.168	40.692	47.859
	FE (L/h = 100)	40.526	38.353	41.011	48.414
	Present study (L/h = 40)	40.998	38.776	41.486	49.098
	Present study (L/h = 100)	41.165	38.913	41.667	49.308
	Yu et al. [16,17]	41.204	38.950	41.706	49.351
$\begin{array}{c} \bar{q}\\ \left(\bar{\omega }=0\right) \end{array}$	FE (L/h = 40)	195.646	87.064	280.818	195.858
	FE (L/h = 100)	203.092	93.052	287.700	203.372
	Present study (L/h = 40)	210.304	99.528	299.924	210.583
	Present study (L/h = 100)	213.167	101.303	305.095	213.468
	Wang et al. [13,14]	213.72	--	306.09	214.02

* FE results calculated with FE code ABAQUS (element type: C3D8R); existing theoretical predictions [13,14,16,17] did not account for shear effect of the monolithic plate and are independent of plate slenderness ratio L/h [10].

## Data Availability

The data presented in this study are available on request from the references and the corresponding author. The data are not publicly available due to the privacy of program data.

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
