# Peer review of "A Three-Dimensional Vibration Theory for Ultralight Cellular Sandwich Plates Subjected to Linearly Varying In-Plane Distributed Loads"

_materials, 2023, doi:10.3390/ma16114086_

Round 1

Reviewer 1 Report

This is a technical paper that establishes a three-dimensional vibration theory for ultralight cellular-cored sandwich plates subjected to linearly varying in-plane distributed loads. The theory accounts for the cross-section rotation angle induced by face sheet shearing and enables the quantification of the influence of core type, boundary conditions, geometric parameters, and face sheet materials on sandwich vibration. The study validates the theory through finite element simulations and finds good agreement between theoretical predictions and simulation results. The paper concludes that the presence of in-plane distributed loads significantly affects fundamental frequencies and modal shapes and that the triangular corrugated sandwich plate possesses the highest fundamental frequency, irrespective of boundary conditions.

I have few questions regarding this paper including:

1.      What are the limitations of the zig-zag displacement model used in this study?

2.      How applicable are the results to sandwich plates with core materials other than those considered in the study?

3.      Can the theory predict the damping behavior of sandwich plates subjected to linearly varying IPDLs?

4.      How do the results of this study compare to previous research on the vibration behavior of sandwich plates in hypergravity environments?

5.      What are the practical implications of the findings for the design and manufacture of ultralight cellular-cored sandwich plates for aerospace applications?

Author Response

  1. What are the limitations of the zig-zag displacement model used in this study?

Answer: The displacement model of sandwich plate adopted in this paper is essentially the superposition of the linear zig-zag function and the first-order shear deformation function of single plate, and no shear correction factor is required. And the in-plane displacement of the reference layer is not considered, which is a classical linear displacement field hypothesis for sandwich structures in Allen’s works. Different from the classical linear displacement filed model which is only applicable to sandwich structures with soft core and hard panel, the displacement model adopted in this study can be applied to sandwich structures with various types of core and panel theoretically. For the rotations ∂w/∂x and ∂w/∂y of the cross-sections are replaced by zig-zag functions θc and θf. The limitation of the present displacement model is that it only considers the linear zig-zag function and may not be suitable for solving the sandwich structure which needs to use the higher-order function to represent the rotation of the cross-section. Especially when a sandwich layer is particularly soft or thick, the shear effect is very obvious. A linear zig-zag function cannot accurately express the shear deformation in this case. In addition, the present displacement model only considers the continuity condition of interlaminar displacement, but does not consider the shear stress, which can be taken into account in the follow study.

  1. How applicable are the results to sandwich plates with core materials other than those considered in the study?

Answer: In present study, three typical core configurations (cellular foam, hexagonal honeycomb and triangular corrugated) are considered. In the case of equal mass, the hexagonal and triangular corrugated cores have higher core thickness than cellular foam core. And compared with hexagonal honeycomb core, triangular corrugated core shows anisotropic mechanical characteristics. In present study, the results show that the vibration characteristics of the sandwich plates are quite sensitive to either core density or core thickness. The higher core thickness leads to the higher fundamental frequency. The mechanical properties of different core materials also have different effects on the vibration behavior of sandwich plates under different boundary conditions. Therefore, based on the present results, the vibration characteristics (such as fundamental frequency and modal shape) of the sandwich plates with other core materials can be qualitatively analyzed in specific situations, which can be used to determine the optimization direction of core materials. As for the specific vibration behaviors, the solution method introduced in this paper can be used for further calculations.

  1. Can the theory predict the damping behavior of sandwich plates subjected to linearly varying IPDLs?

Answer: Theoretically, the present theory can be used to predict the damping behavior of sandwich plates subjected to linearly varying IPDLs as long as the damping effect is considered in the structural stiffness matrix K. However, this work may be a little more complex than the present research, and may be investigated in the future work.

  1. How do the results of this study compare to previous research on the vibration behavior of sandwich plates in hypergravity environments?

Answer: According to the author’s knowledge, the vibration behavior of sandwich plates in hypergravity environments is analyzed in present research, for the first time. Hence, the analysis results in present study are verified by the calculation results of monolithic plate in the literature and the finite element simulations of sandwich plates performed by the authors. The calculated results are in good agreement.

  1. What are the practical implications of the findings for the design and manufacture of ultralight cellular-cored sandwich plates for aerospace applications?

Answer: For the cellular-cored sandwich plates applied in aerospace industry, it is inevitable to bear the in-plane distributed loads (IPDL) induced by hypergravity environments during the process of acceleration for all kinds of aerocrafts. In this process, in order to prevent the excessive dynamic load of the space structure and ensure the normal operation of the spacecraft attitude control system, it is necessary to improve the structural stability and fundamental frequency. Therefore, when the engineers choose the core type based on the multi-functionality of the cellular-cores sandwich plate, the findings of this study can also provide suggestions for them, so that the selected sandwich structure has the maximum structural stability under the application condition. In addition, when considering the use of fiber composite face sheets to further enhance the structural stiffness, the results of this study can provide optimization suggestions for the selection of fiber laying angle. For example, when the fiber laying direction is parallel to the structure acceleration direction, the structure stability is least affected by the IPDL.

Reviewer 2 Report

Review paper: A three-dimensional vibration theory for ultralight cellular sandwich plates subjected to linearly varying distributed loads

The paper presents a comprehensive study on the vibration analysis of ultralight cellular-cored sandwich plates subjected to linearly varying in-plane distributed loads (IPDL), with the transverse shear effect of face sheets accounted for. The study establishes a three-dimensional (3D) vibration theory based on the zig-zag displacement model, and validated through 3D finite element simulations. The influence of core type, geometric parameters, and face sheet materials on sandwich vibration is systematically quantified, with a particular focus on the fundamental frequency and critical buckling IPDL.

However, after being carefully reviewed, the manuscript is interesting to the readership and fits well into the Materials journal. However, it needs some improvement for publication in Aerospace. The authors should address the following points:

1.     The significance of current research work is unclear. The significance of using the zig-zag displacement model is not clearly explained in the paper, and the authors should provide a more detailed review and explanation of its use.

The author can find the detailed information below recent references:

§  Sayyad, A.S. and Ghugal, Y.M., 2017. Bending, buckling and free vibration of laminated composite and sandwich beams: A critical review of literature. Composite Structures, 171, pp.486-504.

§  Kutlu, A., Dorduncu, M. and Rabczuk, T., 2021. A novel mixed finite element formulation based on the refined zigzag theory for the stress analysis of laminated composite plates. Composite Structures, 267, p.113886.

§  Nguyen, S.N., Lee, J. and Cho, M., 2016. A triangular finite element using Laplace transform for viscoelastic laminated composite plates based on efficient higher-order zigzag theory. Composite Structures, 155, pp.223-244.

2.     Eq. 5: Please note the order of the stress and strain. It is still correct, but not in standard order. It makes the reader confuse a lot, even for the following formulation.

3.     Please check detail typos and their format, especially the mathematical parameters.

4.     Appendix B: Please give the details of the position inside the stiffness matrix K.

In Results and Discussion

5.     The authors should provide a more detailed explanation of the boundary conditions considered, maybe give one more picture with BC and linearly varying distributed loads. Furthermore, explain in more detail how they affect the vibration modal shapes.

6.     The authors should consider providing some insights into the physical mechanisms that govern the influence of core density or core thickness on the fundamental frequency and critical buckling IPDL.

Minor editing of English language required

Author Response

  1. The significance of current research work is unclear. The significance of using the zig-zag displacement model is not clearly explained in the paper, and the authors should provide a more detailed review and explanation of its use. The author can find the detailed information below recent references:

Sayyad, A.S. and Ghugal, Y.M., 2017. Bending, buckling and free vibration of laminated composite and sandwich beams: A critical review of literature. Composite Structures, 171, pp. 486-504.

Kutlu, A., Dorduncu, M. and Rabczuk, T., 2021. A novel mixed finite element formulation based on the refined zigzag theory for the stress analysis of laminated composite plates. Composite Structures, 267, p. 113886.

Nguyen, S.N., Lee, J. and Cho, M., 2016. A triangular finite element using Laplace transform for viscoelastic laminated composite plates based on efficient higher-order zigzag theory. Composite Structures, 155, pp.223-244.

Answer: The highlights of current research work is that the vibration performance of cellular-core sandwich plates subjected to linearly varying in-plane distributed loads (IPDLs) is firstly analyzed, with the transverse shear effect of face sheets accounted for. Before this work, the influence of IPDLs and face sheets’ shear effect on the vibration analysis of the sandwich plates are not simultaneously analyzed. Our results proved the necessity of considering the influence in some cases. In addition, as suggested by the reviewer, the zig-zag displacement model is reviewed and explained in more detail in the introduction. The revision has been marked in the revised draft.

  1. Eq. 5: Please note the order of the stress and strain. It is still correct, but not in standard order. It makes the reader confuse a lot, even for the following formulation.

Answer: The order of the stress and strain in Eqs. 5 and 6 is modified, and the subsequent equations are checked. The revision has been marked in the revised draft.

  1. Please check detail typos and their format, especially the mathematical parameters.

Answer: The detail typos and formats have been checked in the draft carefully, and the inappropriate writing has been modified.

  1. Appendix B: Please give the details of the position inside the stiffness matrix K.

Answer: The details of the stiffness matrix K has been added. The revision has been marked in the revised draft.

In Results and Discussion

  1. The authors should provide a more detailed explanation of the boundary conditions considered, maybe give one more picture with BC and linearly varying distributed loads. Furthermore, explain in more detail how they affect the vibration modal shapes.

Answer: A more detailed explanation of the boundary conditions considered has been added in Fig. 3 and Table 1. And a more detailed description among the relationship of BC, IPDLs and modal shapes has been added in section 4.5. The revision has been marked in the revised draft.

  1. The authors should consider providing some insights into the physical mechanisms that govern the influence of core density core thickness on the fundamental frequency and critical buckling IPDL.

Answer: A more detailed discussion about the physical mechanisms that govern the influence of core density and core thickness on the fundamental frequency and critical buckling IPDL has been added in section 4.4. The revision has been marked in the revised draft.

Reviewer 3 Report

Undertaken problem of sandwich plates with different geometric (material, too) parameters of structure under natural loads is very interesting, practically and scientifically valuable.

I recommend to accept it for publishing but after some corrections and supplements:

1)      In point 1 of manuscript there are many references to literature. I propose to emphasize the main, new elements ow work.

2)      The Fig. 1 part b and c should be magnified to show  better the notation of structure geometry.

3)      The commas should be checked. Also, the greek letter “ni”. Should be ν.

4)      The reference to classic sandwich theory with zig-zag cross-section presented for example in work by Wolmir, A.S. should be presented. There is distribution of normal and shear stresses to facings and core.

5)      The main parameters should be presented I added notation.

Author Response

  1. In point 1 of manuscript there are many references to literature. I propose to emphasize the main, new elements ow work.

Answer: The references in point 1 has been streamlined. And the main, new elements ow work have been emphasized. The revision in point 1 has been marked in the revised draft.

  1. The Fig.1 part b and c should be magnified to show better the notation of structure geometry.

Answer: The Fig. 1 has been modified and the subscripts have been magnified. Please review that in the revised version.

  1. The commas should be checked. Also, the greek letter “ni”. Should be ν.

Answer: The commas have been checked. And the Greek letter of Poisson’s ratio have been modified to ν in the revised draft.

  1. The reference to classic sandwich theory with zig-zag cross-section presented for example in work by Wolmir, A.S. should be presented. There is distribution of normal and shear stresses to facings and core.

Answer: The work by Volmir, A.S. has been added to the reference, such as ‘Volmir, A.S.: “Stability of sandwich plates and shells”, Stability of Deformable S., Nauka Publ House, Moskow(in Russian), Ch.19, 1967’. Please review that in the revised version.

  1. The main parameters should be presented I added notation.

Answer: The main parameters refer to the core thickness , core density , corrugation angle , and etc.). The illustration has been added to the revised draft.

Round 2

Reviewer 1 Report

The authors have addressed all the comments. It may be accepted for publication.